# Unicorns, Rhinoceroses and Chemical Bonds

**DOI:** 10.3390/molecules28041746

**Published:** 2023-02-12

**Authors:** Jordan Gribben, Timothy R. Wilson, Mark E. Eberhart

**Affiliations:** 1Chemistry Department, Loras College, 1450 Alta Vista Street, Dubuque, IA 52001, USA; 2Chemistry Department, Colorado School of Mines, 1500 Illinois Street, Golden, CO 80401, USA

**Keywords:** bond bundle, bond energy, bond analysis, electron density, Jahn–Teller, FCC, QTAIM

## Abstract

The nascent field of computationally aided molecular design will be built around the
ability to make computation useful to synthetic chemists who draw on their empirically based
chemical intuition to synthesize new and useful molecules. This fact poses a dilemma, as much
of existing chemical intuition is framed in the language of chemical bonds, which are pictured
as possessing physical properties. Unfortunately, it has been posited that calculating these bond
properties is impossible because chemical bonds do not exist. For much of the computationalchemistry
community, bonds are seen as mythical—the unicorns of the chemical world. Here, we
show that this is not the case. Using the same formalism and concepts that illuminated the atoms in
molecules, we shine light on the bonds that connect them. The real space analogue of the chemical
bond becomes the bond bundle in an extended quantum theory of atoms in molecules (QTAIM).
We show that bond bundles possess all the properties typically associated with chemical bonds,
including an energy and electron count. In addition, bond bundles are characterized by a number of
nontraditional attributes, including, significantly, a boundary. We show, with examples drawn from
solid state and molecular chemistry, that the calculated properties of bond bundles are consistent
with those that nourish chemical intuition. We go further, however, and show that bond bundles
provide new and quantifiable insights into the structure and properties of molecules and materials.

## 1. Introduction: History of the Chemical Bond

Just as computer-aided design transformed the way things of the modern world—from microscopic semiconductor devices to kilometer-long bridges—are developed, the emerging field of molecular design will alter the manner through which the atomic-scale structures composing these things are discovered, improved, and utilized. Already, this nascent field is showing promise by accelerating advances in pharmaceuticals, artificial enzymes, catalysts, and materials [1,2,3,4,5].

In part, molecular design’s potential will be realized by merging the computational and synthetic branches of chemistry into a single subdiscipline. Herein lies a great challenge. On the one hand, modern computational chemistry relies on the ever-expanding power of computers and computational methods to determine with near-chemical accuracy the thermodynamic properties of molecular systems. On the other hand, synthetic chemistry draws in large part on its practitioners’ *chemical intuition*—the term used to refer collectively to a vast and growing empirical knowledge base of chemical reactions that are rationalized with an intricate set of heuristic models underpinned by the *chemical bond*.

The engaging review, “Chemical Bonding: The Journey from Miniature Hooks to Density Functional Theory” by Constable and Housecroft [6] traces the progress in our understanding of the chemical bond from the Greek atomists to its several modern representations. The story, though, is less one of a relentless march toward fundamental understanding and more a recounting of an evolutionary process through which modern bonding theories diverged from a common centuries-old ancestor via selective empirical pressures. The emerging models are adapted to specific environments: one model for organic molecules, another for coordination compounds, still another for the solid state, etc.

Taken together, these models give rise to what Hoffmann described as the “fuzzy richness” of the bond [7]. Its chimeric properties prompted Frenking and Krapp [8] to compare chemical bonds to unicorns, “mythical but useful creatures that bring law and order to an otherwise chaotic and disordered world.” In addition, though chemists speak of the bond and evoke its properties, like a unicorn, no one has actually seen one. Bader, rejecting mythological allusions, stated the case simply [9]; “a ‘bond’ is neither measurable nor susceptible to theoretical definition”. Coulson had a similar thought, commenting [10], “Sometimes it seems to me that a bond between two atoms has become so real, so tangible, so friendly, that I can almost see it. Then I awake with a little shock, for a chemical bond is not a real thing. It does not exist. No one has ever seen one. No one ever can. It is a figment of our own imagination”.

Yet, it is the ability to manipulate this fuzzy, mythical, imaginary bond that—in what is as much art as science—allows synthetic chemists to create their molecular masterpieces.

How, then, can computation be used to give substance to the chemical bond and enhance chemical intuition? One approach is through computational empiricism, where energy and other thermodynamic properties of simulated molecules and reactions are determined and then interpreted from the most appropriate bonding vantage point. This method is made all the more useful by the ability to computationally study systematically altered molecules and reactions to include those that do not exist and, hence, cannot be studied experimentally. The trends emerging from such investigations provide invaluable data from which to tune existing bonding models and, in some cases, to predict the properties of molecules and substances that have yet to be made. Taken to larger scales, machine-learning and artificial-intelligence algorithms provide the upmost combination of computational empiricism and statistical analysis, and are increasingly used to better understand and predict molecular and material properties [11,12,13,14,15,16,17].

Still, we cannot help but note that this epistemology is similar to finding evidence of unicorns in narwhal tusks, or justifying the Ptolemaic model of the universe by the “discovery” of new epicycles.

Is it possible that the chemical-bond concept only appears fuzzy because what we see using our current tools is but the shadow cast by a dimly lit structure? If we illuminate the object with the full intensity of modern computation, could we then see the sharp edges, corners and surfaces that are the origin of its properties? Here, we will provide evidence to support this conjecture.

We will build on earlier work showing that what we call a “bond bundle” possesses all the presumed properties of the real-space bond, including: a specific number of electrons, an energy and, hence, energy-related properties, and, importantly, a boundary separating bond bundles and through which bond–bond interactions are mediated. After developing the concepts, we include two examples using one solid state and one molecular system; pure silver and cyclobutadiene, respectively. The two provide contrast for organic vs metallic bond bundles, while the cyclobutadiene example dissects a familiar chemical phenomenon, the Jahn–Teller distortion, including the assessment of individual bond energies.

## 2. Bond Properties

We begin not by assigning to an observed structure a set of bond-like properties (e.g., a molecular geometry, a bond point or path), but by identifying the essential properties of a bond and then searching for structures possessing these properties. In turn, a bond’s essential properties are those necessary to rationalize a molecule’s stability as arising from the general form of its electronic structure.

The set of essential bond properties is quite compact. They are: (1) a bond contains a specific number of electrons—its electron count; (2) an electron is contained in one and only one bond; (3) a bond possesses an energy; and, (4) bond energies may be added to recover total molecular energy. Together with our desire that bonds be physical observables, properties 1 and 2 demand that bonds occupy real space and are characterized by non-overlapping bond volumes. As a corollary, then, bonds must be bounded by explicit surfaces. Properties 2 and 4 require that bond volumes fill space.

### QTAIM and Kinetic Energy Ambiguity

Properties 1 and 2 are among the most useful of modern chemistry, with the inter-bond electron flow accompanying chemical reactions often described with curved arrows [18]. Despite the fact that such representations mandate the existence of surfaces separating one bond from another, there has been little curiosity concerning the location and structure of these surfaces. The reason for the dearth of interest may stem from the apparent intractability of the problem. There are, after all, an infinite number of surfaces that could be chosen as bond boundaries, and without a compelling rationale, the choice of any particular boundary is arbitrary.

The same conundrum confronted efforts to find the boundaries separating atoms in molecules. Bader, in his quantum theory of atoms in molecules (QTAIM) [19], solved the problem by requiring such atoms to have unambiguous energies. As there are multiple equally valid kinetic-energy operators, in particular the gradient and Laplacian operators (${\hat{T}}_{G}$ and ${\hat{T}}_{K}$, respectively) [20,21,22], the value of the local kinetic energy cannot be determined with specificity [23]. However, over regions bounded by surfaces of zero flux in the gradient of the charge density, $T_{G}=T_{K}$, a regional kinetic energy was argued to be well-characterized. As energy is a desired property for atoms in molecules, Bader specified that these atoms should be bounded by the unique surface containing a single nucleus and over which the flux of the gradient of the charge density is everywhere zero, a so-called zero-flux surface (ZFS). The volume bounded by these surfaces is called an atomic basin or a Bader atom, which, by construction, is characterized by an explicit kinetic and, hence, total energy.

Others have noted [24,25] that there are alternative boundaries that would also yield precise regional energies. Therefore, they have argued, there is no *a-priori* reason to choose the ZFS condition over others. However, such arguments obscure a QTAIM strength: the revelation of structure–property relationships wherein atomic properties such as electron count, energy, and energy-related properties are a consequence of the structure of ZFSs characterized by computable attributes such as shape, volume, local and total curvature, etc. Having established the existence of these relationships, the utility of QTAIM as a design tool reduces to providing a scientific rationale for these relationships, then demonstrating an ability to chemically manipulate the structures mediating desired properties. Choosing an alternative boundary would simply recover another set of structure–property relationships. The “better” choice is the one for which existing chemical intuition is more compatible.

#### The Space of All ZFS and Condensed Properties

That the structure of a Bader atom’s boundary and its energetic properties are related is but a single facet of a larger set of relationships through which molecular properties are controlled by the twin structures of the charge-density gradient field, $\nabla \rho (\vec{r})$, and its dual representation as charge-density isosurfaces.

As has been discussed previously [26,27,28,29,30,31,32,33], we picture $\nabla \rho (\vec{r})$ as a set of arc-length-parameterized gradient paths (G) originating from a local charge-density minimum—a cage critical point (CP)—and terminating at a maximum—almost always coincident with a nuclear CP. Imagine every nuclear CP as the center of a reference sphere (S${}_{i}$) of radius dr with a G passing through all the points on its surface. In a familiar way, the points on such a sphere may be specified by a polar and an azimuthal angle, allowing each of the molecule’s Gs to be referenced by a pair of coordinates and the index of the nuclear CP at its terminus, i.e., G${}_{i}\left(\theta ,\phi \right)$.

If each S${}_{i}$ is covered by a set of non-intersecting differential elements of area dA, then the Gs passing through the points interior to these area elements generate a family of infinitesimal volume elements called differential gradient bundles (*d*GB${}_{i}\left(\theta ,\phi \right)$) [30,31], whose cross-sectional areas change along their length. As each *d*GB${}_{i}\left(\theta ,\phi \right)$ is, by construction, bounded by a ZFS, the integral of the total energy over these volume elements will yield an explicit regional value called the condensed total energy, $E_{i}\left(\theta ,\phi \right)$.

In an equivalent manner, for any 3D scalar field, *f*, there exists a corresponding 2D condensed scalar field for each nuclear CP, $F_{i}$, which is a function of θ and ϕ and a functional of *f*, where
(1)$$F_{i}\left[f\right]\equiv F_{i}\left[\theta ,\phi ,f\left(\theta ,\phi ,s\right)\right]=\;\;\;\int_{G_{i}\left(\theta ,\phi \right)}\;\;\;f\left(s\right)\;dA\left(s\right)\;ds.$$
That is, the 2D condensed scalar field value at the angle (θ,ϕ) results from the path integration along the arc length (*s*) of $G_{i}\left(\theta ,\phi \right)$, to which *d*GB${}_{i}\left(\theta ,\phi \right)$ converges at the nuclear CP. In particular, the charge density yields the condensed charge density (F[ρ]=P), the Laplacian or gradient kinetic-energy densities yield the condensed kinetic-energy density ($F\left[T_{G}\right]=F\left[T_{K}\right]=T$), and so forth. For the case where f(θ,ϕ,s)=1, the differential gradient bundle volume results (F[1]=V). Visual representations of condensed properties are achieved by mapping these values onto S${}_{i}$. By way of illustration, we consider two examples, a solid state and a molecular system.

## 3. Computational Details

All calculations were performed using the software from Amsterdam Modeling Suite. The BAND package [34,35,36] was used to compute the charge densities and energies of the solid-state example while the Amsterdam density functional (ADF) method [37,38] was used to compute the charge densities and energies of the molecular systems. In both cases, an all-electron TZ2P basis set with a GGA-PBE exchange correlation functional was employed.

Condensed charge densities and energies were determined using our in-house Bondalyzer package [39,40,41] using Tecplot 360 for graphics and visualization [42].

Total energies reported are those exploiting the virial theorem [19,23,43], where for a system for which no forces are acting on the nuclei, the average kinetic energy over a well-defined region, $T_{\Omega }$, equals the negative of the region’s total energy, i.e., $E_{\Omega }=-T_{\Omega }$. For our purposes, for $T_{\Omega }$ we use the non-interacting kinetic energy, which omits kinetic energy due to exchange and correlation, $T_{c}$. The extent of the error introduced through this approximation can be determined from the ratio of the system potential energy to the total non-interacting energy, −VT. Deviations from 2 indicate the extent to which the non-interacting kinetic energy under estimates the total kinetic energy. This is typically a small number and in the case of the calculations presented here the virial ratio was found to be between 2.003 and 2.005 indicating an approximation error of between 0.3 and 0.5% to the values of the total energies [44].

## 4. Bond-Wedges and Bond Bundles

Beginning with an example from the solid state, the top right pane of Figure 1 is meant to represent several of the *d*GBs of FCC silver. The bottom panes show the condensed total energy (E) and charge density (P) distributions, respectively.

As every point on S${}_{i}$ maps to a G, every trajectory on S${}_{i}$ maps to a ZFS of ρ, and any closed loop maps to a ZFS bounded volume with an unambiguous energy. Such volumes are called gradient bundles.

Just as all Gs terminating at the same maximum in ρ delineate an atomic basin as a unique volume, we define a set of condensed property basins as the set of condensed property gradient paths on each S${}_{i}$ that terminate at the same condensed property maximum or minimum. This gives rise to a set of loops in condensed space through which the flux of a condensed property (e.g., E, P) is zero. These loops are the images in condensed space of unique gradient bundles, which may be thought of as the the volume in 3D over which an atom’s property is concentrated.

The basin in which P is concentrated has been designated a bond wedge [33]. A bond bundle is defined to be the union of bond wedges sharing a common intersection with an interatomic surface. Figure 2 shows this structure in crystalline Ag.

Though this is not the case for all FCC transition metals, Ag is characterized by a single type of bond wedge, of which there are six for each Ag atom, as shown in Figure 2. This large bond wedge shares intersections with the identical bond wedges from each of the six neighboring atoms at the vertices of the FCC octahedral hole—forming a 6-center bond bundle with a QTAIM cage-CP at its center and bond paths along its faces.

As bond wedges and bundles are bounded by ZFSs, each has a well-defined energy. The total energy of a bond wedge in Ag is found to be −1052.71 Ha, which, by symmetry, is one sixth of the Ag atom energy.

Again, though this not typical of all FCC metals, there is also one type of condensed energy basin characterizing FCC Ag (Figure 1). As there is only one type of bond wedge and energy basin, by symmetry the boundaries of the two must coincide. However, even in cases where symmetry does not demand it, we have found that the boundaries of the ZFSs of the P and E basins lie very close to each other, and in organic molecules they are visually coincident. As the approximations we use to calculate E are less precise than those used to determine P, we cannot rule out the possibility that bond-bundle ZFSs are coincident with the ZFSs bounding E basins. Regardless, it appears that ZFS bounded regions over which ρ is concentrated are also regions which contribute to the stability of the system by minimizing total energy.

In addition to the various named gradient bundles and their integrated properties, each possesses a number of geometric properties. One of these is its solid angle, α. The solid angle of a bond wedge—or that of any condensed property basin—is given by the fraction of the reference sphere’s surface it covers. For example, α for the E basins of Ag is 16. As another example, α for the C and H bond-wedges of methane or ethane are 14 and 1, respectively. In benzene, α for the C bond wedge of the C–C bond bundle is 0.35 and for the C bond wedge of the C–H bond bundle it is 0.30. In ethylene, the C bond-wedge components of the C=C and C–H bond bundles have α of 0.42 and 0.29. In acetylene, α for a C bond wedge in the triple bond is 0.62, with the remaining 0.38 going to the C–H bond bundle. These prototypical molecular bond wedges and bond bundles are depicted in Figure 3. An important takeaway is that the solid angle is a parameter akin to hybridization, but more sensitive.

With the solid-state example as background, we turn now to a more substantive example by determining the changes in the bond bundles accompanying the D${}_{4h}$ to D${}_{2h}$ distortion of cyclobutadiene.

Cyclobutadiene is a highly reactive antiaromatic molecule that, due to its tendency to dimerize, has not been isolated in its parent form, though some substituted derivatives have been found [45,46]. Despite these facts, it is well-studied theoretically as a simple example of a Jahn–Teller system [47]. In the lowest energy, spin-restricted, D${}_{4h}$ configuration, all C–C distances are 143.3 pm and there is a symmetry required HOMO-LUMO degeneracy. Upon symmetry lowering to D${}_{2h}$, the degeneracy is removed as one pair of opposing C–C distances lengthens to 156.3 pm, while the other set shortens to 133.0 pm. The ground-state D${}_{2h}$ geometry is computed to be approximately 0.6 eV more stable than the triplet and about 1.0 eV more stable than the spin-restricted square-planar geometry.

Figure 4 depicts the dramatic qualitative changes in the bond bundles of cyclobutadiene accompanying the D${}_{4h}$ to D${}_{2h}$ distortion. The corresponding quantitative changes to the total charge, the energy, and the solid angle of the molecule’s four distinct bond wedges are given in Table 1. In addition to the bond wedge on H that is part of the C–H bond bundle (H of C–H), there are three bond wedges on C: one that is part of the aforementioned C–H bond bundle (C of C–H), another that is part of the developing C=C bond bundle (C of C=C), and one that is part of the developing C–C bond bundle (C of C–C). In the D${}_{4h}$ configuration, the latter two bond wedges are symmetry-equivalent.

A salient feature is the bond-wedge energies and their changes. Recall that these are the energies of all the electrons in the bond wedge, which necessarily includes the “core” electrons. The distinction between core and valence is not strictly possible in a model based on the observable total density. However, various approximations can be made that will allow for this separation, as we will see. For now, though, we are interested in the magnitude of the changes through symmetry breaking.

The changes to the solid angles and electron counts accompanying the distortion are worth noting. In the high-symmetry configuration, the solid angles are those of an $sp^{2}$-type C, with α for each of its three bond wedges close to 13. Upon symmetry lowering, the solid angle of the C bond wedge that is part of the developing double bond increases to 0.41 (very close to the value of the corresponding bond wedge in ethene) while that for the developing single bond decreases to 0.27, consistent with an $sp^{3}$ C. At the same time, there is a transfer of approximately half an electron from the bond wedge to the developing single bond bundle to the bond wedge of the developing double bond bundle.

The total energy of the C–H fragment obtained by summing the wedge energies recorded in Table 1 yields the D${}_{4h}$ and D${}_{2h}$ configuration energies of −38.416 and −38.425 Ha, respectively. The stabilization energy from the distortion is, accordingly, 0.009 Ha per C–H fragment or 0.036 Ha per molecule, equivalent to approximately 1.0 eV, which is 2% lower than the computed stabilization determined from ADF calculations of total molecular energy. There is nothing surprising here, as, clearly, summing the energy of ZFS bounded regions must recover the total energy of the system and, hence, distortion energies. However, what is surprising, and certainly novel, is the way the stabilization energy is distributed among the bond bundles of D${}_{2h}$ cyclobutadiene.

Before proceeding with this analysis, we will place the data of Table 1 into the more traditional framework given in Table 2, where the number of valence electrons in each of the three bond bundles and their energies, in kJ/mole, relative to a standard state are reported.

The number of valence electrons in each bond bundle was found by subtracting off the core electrons contained in each wedge. In turn, the number of wedge core electrons is given by the product of the solid angle spanned by the bond wedge with the number of core electrons in the isolated atom. The valence bond-bundle electron count is then given as the sum of the valence electrons in the wedges comprising the bond bundle.

The bond energy, that is, the energies relative to an isolated atom reference state, was computed in a similar fashion. By way of illustration: The energy of an isolated C atom, E${}_{C}$, was computed to be 37.749 Ha. Thus, the atomic contribution to the energy of a C bond wedge is given by the product of the bond-wedge solid angle with E${}_{C}$. This quantity is subtracted from the particular bond-wedge energy of Table 1 to give the bond energies of Table 2. That is, the bond-bundle energy difference between the isolated atomic and molecular states.

The data of Table 2 is reasonable but differs from traditional views in several respects. First, the valence electron count of the C–H bond bundle is a bit higher and that of the D${}_{2h}$ C=C bond bundle is lower than conventional expectations based on the Lewis model. The bond energies of the D${}_{4h}$ system are again reasonable with the lower C–H bond-bundle energy, and the C–C bond-bundle energies are a little larger than the conventional view (413 and 480 kJ/mole, respectively). However, the conventional values are based on a number of approximations, not the least of which is that these bond energies are determined by averaging thermodynamic and theoretical data across many systems without any clear methodology to partition energy and density between the bonds. Though the precision with which bond-bundle energies are determined can be improved, the methodology is sound and will lead to unequivocal energies based on the structure of the charge density.

With this observation in mind, the change in bond energies through the distortion is dramatic and, for us, unexpected. The C=C bond-bundle energy is nearly 75% larger than the conventional C=C bond energy and the C–C bond-bundle energy is only 28% of that expected for a C–C bond. The origin of this behavior can be found in the solid angles. For the single bond bundle, the solid angle of the participating C atoms is that of an $sp^{3}$ C, reducing the volume over which the bond-bundle energy is integrated. In effect, this reduced value of α reflects the ring strain of this four-membered ring, the energy of which is accommodated entirely by the single bond.

From an orbital perspective, in the square-planar geometry, the in-plane carbon *p* orbitals can participate in σ interactions in one direction and, simultaneously, in π interactions in the perpendicular direction. Through the D${}_{4h}$ to D${}_{2h}$ distortion, both the σ and π interactions along the direction in which the C–C distance is shortening intensify and, by virtue of a larger α, are incorporated disproportionally into the C=C bond bundle. Energetically, we may think of cyclobutadiene as a weakly bound and highly strained (∼1 eV) ethyne dimer, a perspective that accounts for its extreme reactivity.

In summary, we showed that there are ZFS bounded volumes called bond bundles which have the properties typically associated with real-space chemical bonds, e.g., energies and electron counts. In addition, bond bundles may be characterized with a host of descriptors that have not previously been considered bond attributes; most importantly, they have boundaries and, hence, possess a number of shape-related characteristics, e.g., solid angles. The shape of a bond bundles is determined solely from the electron density and, hence, the concept is equally applicable to all electronic matter. There are multi-center bond bundles in metals and Lewis-like bond bundles in organic molecules. Accordingly, the bond-bundle concept will allow us to extend our chemical intuition beyond the molecular classes to which it has been finely tuned to all types molecules and solids.

It is thought that the unicorn myth originated in the attempts of ancient travelers from India and Africa to describe rhinoceroses. Having no point for comparison, Europeans elaborated on this description as it grew into the mythical unicorn. Marco Polo, upon seeing an Indian rhinoceros, described what he thought was a unicorn as, “[A] hideous beast to look at, and in no way like what we think and say in our countries”, [48] a description that is unfair to the powerful and majestic rhinoceros, a keystone species that helps to shape their ecosystem. We have seen that the bond is not a mythical unicorn but a rhinoceros, a tangible part of the molecular ecosystem.

## Figures and Tables

**Figure 1 molecules-28-01746-f001:**
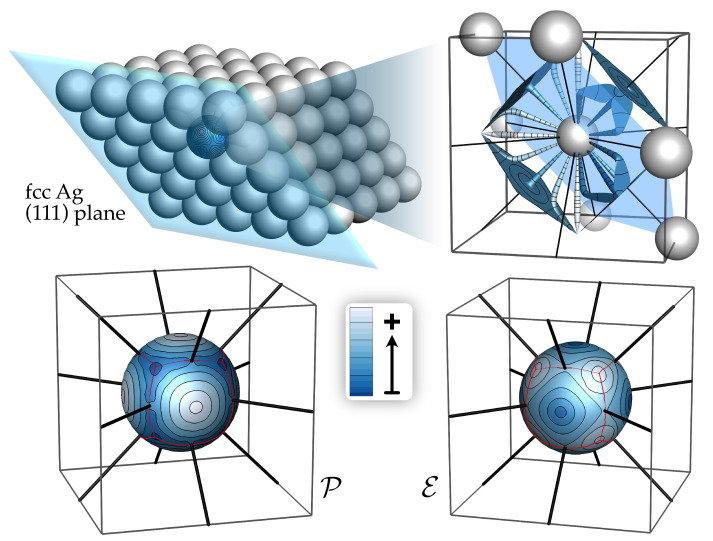
Top right, a sampling of differential gradient bundles tracing the shape of ρ in crystalline silver, lined with contours of ρ and colored according to the amount of ρ contained in each. Integrating the total energy and charge density in all such differential gradient bundles yields the 2D condensed total energy (E) and charge density (P) distributions.

**Figure 2 molecules-28-01746-f002:**
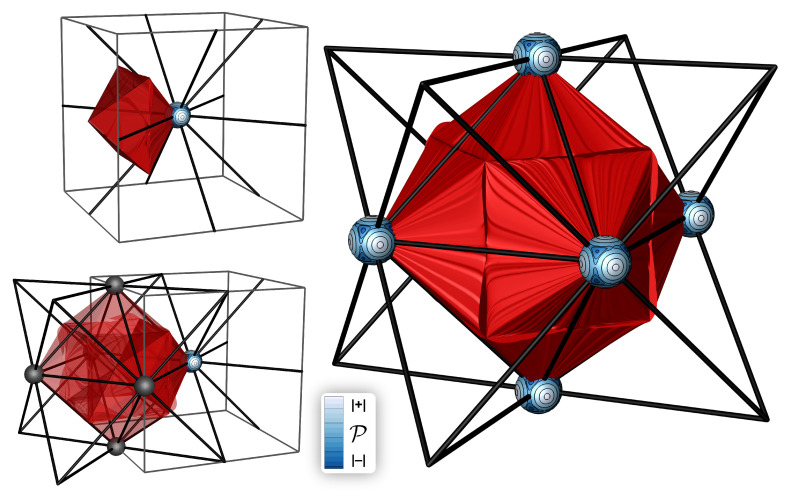
One Ag bond wedge is shown alone at top left, which combines to form a six-center bond bundle which coincides with the octahedral hole.

**Figure 3 molecules-28-01746-f003:**
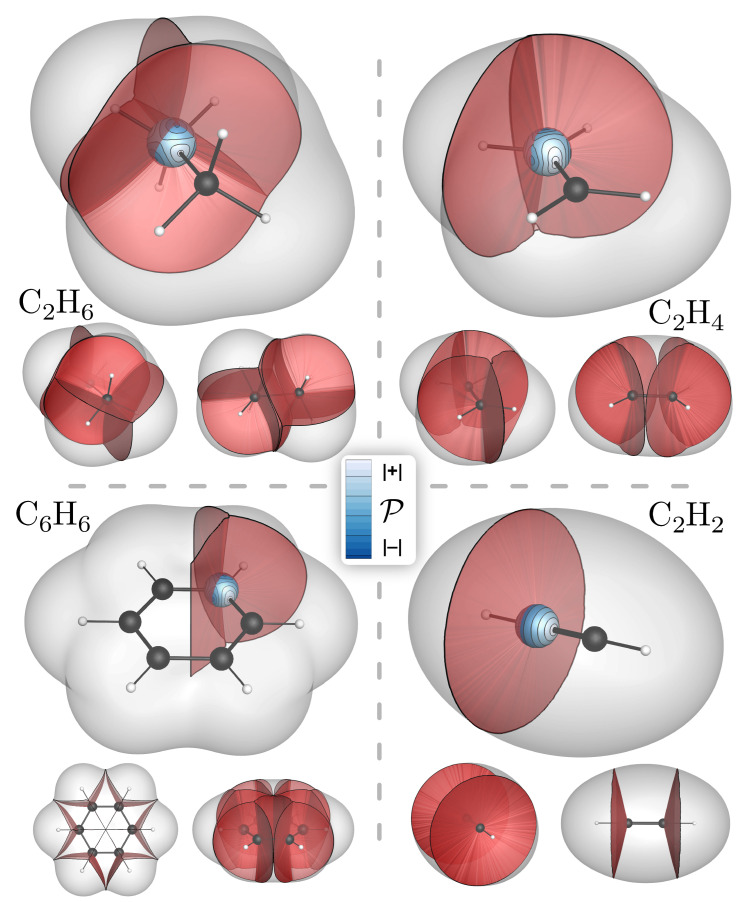
C atom bond wedges shown intersecting reference spheres with mapped contours of P in ethane, benzene, ethylene, and acetylene (bond orders of 1, 1.5, 2, and 3). For each molecule, the full set of inter-bond surfaces is shown. Surfaces are truncated at the ρ=0.001 isosurface.

**Figure 4 molecules-28-01746-f004:**
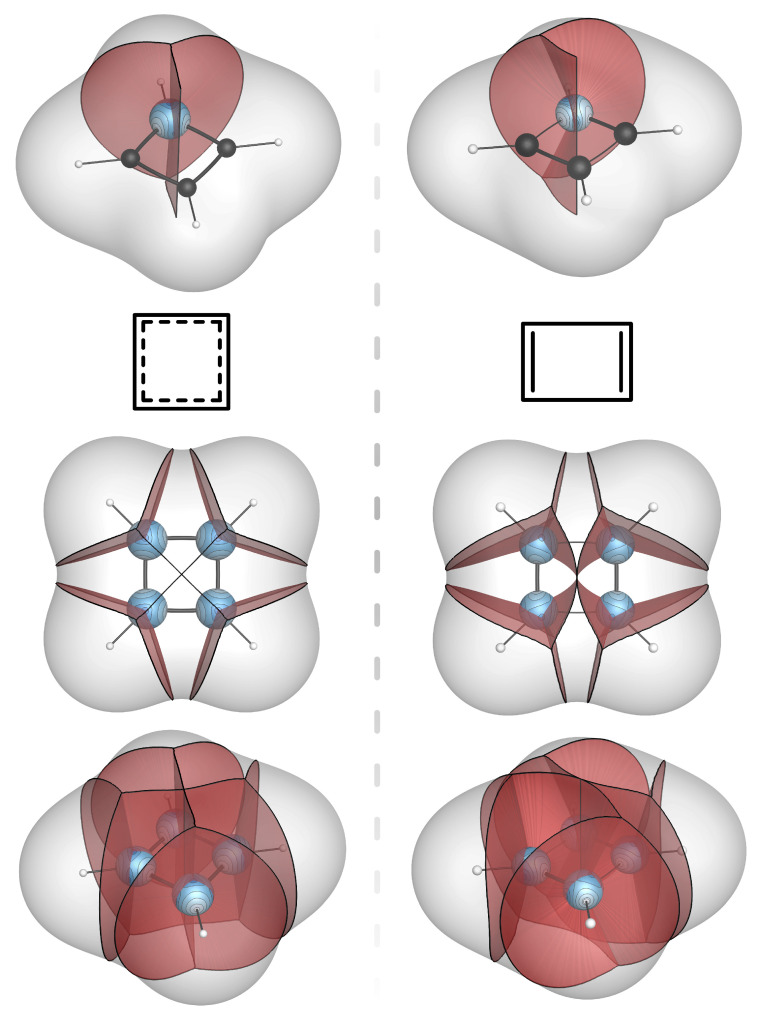
Bond-wedge and bond-bundle surfaces in cyclobutadiene in square-planer and rectangular geometries, intersecting C-atom-centered reference spheres with mapped contours of P. Bond-bundle surfaces are truncated at the ρ=0.001 isosurface.

**Table 1 molecules-28-01746-t001:** Bond-wedge properties.

	ρ [*e*]		−E [Ha]		α	
	D${}_{4h}\to$ D${}_{2h}$		D${}_{4h}\to$ D${}_{2h}$		D${}_{4h}\to$ D${}_{2h}$	
C of C=C	1.991	2.521	12.474	15.506	0.329	0.407
C of C–C	1.991	1.546	12.474	10.043	0.329	0.267
C of C–H	2.053	1.968	12.916	12.325	0.343	0.327
H of C–H	0.924	0.922	0.553	0.552	1	1

**Table 2 molecules-28-01746-t002:** Relative valence electron counts and bond energies.

	Valence		Bond Energy	
	Electrons		[kJ/mol]	
	D${}_{4h}\to$ D${}_{2h}$		D${}_{4h}\to$ D${}_{2h}$	
C=C	2.668	3.416	597.9	1074.9
C–C	2.668	2.027	597.9	98.3
C–H	2.291	2.237	305.8	331.4

## Data Availability

The data presented in this study are in the text and Appendix A.

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
