# Peer review of "Unicorns, Rhinoceroses and Chemical Bonds"

_molecules, 2023, doi:10.3390/molecules28041746_

Round 1
Reviewer 1 Report
The paper from Eberhart and co-workers which theoretically shows that bond-bundles can provide new and quantifiable insights into the structure and properties of molecules and solid-state compounds is a good work which was competently carried out and which can be published in Molecules. I have just a few minor matters that the authors should consider before publication proceeds.
a) Introduction.
1- In addition to ‘merging the computational and synthetic branches of chemistry into a single subdiscipline’ to realizemolecular design, AI may also be part of the story. This could be mentioned and referenced.
2- A few bond descriptors in the real space have been developed and used over the years, see J. Chem. Theory Comput. 2014, 10, 3745 and refs therein for instance. They should be referenced and discussed with respect to the approach used in the paper.
b) The choice of silver as a solid-state compound – I prefer the word ‘solid-state compounds’ rather than ‘materials’ which connotates some application – and cyclobutadiene as examples should be justified.
c) Typo.
Page 10, line 314. thought
Author Response
Responses in italics.
Some minor corrections:
- line 143: provide an explanation for the symbol "s", please. Similarly, the meaning of its differential "ds" could be given below the equation.
- The arclength (s) had been defined two paragraphs above its first use. That has been moved to after eqn 1 for clarity, along with a verbal interpretation of the equation: “the 2D condensed scalar field value at the angle (results from the path integration along the arc length (s) of to which converges at the nuclear CP.”
- line 394: the format of this reference seems somewhat strange.
- This reference is a book chapter in a book with multiple editors. I believe the formatting is correct. Thank you.
Reviewer 2 Report
This work is an interesting attempt to give light on the concept and representation of chemical bonds. Synthetic chemists reason on the basis of chemical intuition and speak about chemical bonds. Computational chemists think that chemical bond is a primitive artifact. This dichotomy suggests the need of more reflection to link both worlds. Authors identify bonds with bond-bundles (related to bond-wedges with particular solid angles) in an extended quantum theory of atoms in molecules. They carefully apply their model to a silver atom in the FCC face of the silver network and to the symmetry lowering from D4h to D2h of cyclobutadiene. These examples allow to see how the model works and how it compares with other well-stablished models. Some surprising results are found for the second example, but are also somehow rationalized. The study is rigorous and authors do not hide aspects that need improvement but, as a whole, the work is interesting and deserves to be published.
A small correction is suggested:
In the Introduction, Subtitle 1.1. History of the Chemical Bond is not necessary, since there are no other section in the introduction one.
Author Response
- In the Introduction, Subtitle 1.1. History of the Chemical Bond is not necessary, since there are no other section in the introduction one.
- Thank you.
Reviewer 3 Report
In the manuscript the Authors present an extension to the well-known QTAIM theory. Within this extention, they proposed two new concepts: the bond-bundle and the bond-wedge. The applicability of these concepts was proved for several molecular examples and a single solid-state structure. The manuscript is well prepared and might be interesting to the readers of the journal. In my opinion, the manuscript in its current form could actually be published 'as is'.
Some minor corrections:
1. line 143: provide an explanation for the symbol "s", please. Similarly, the meaning of its differential "ds" could be given below the equation.
2. line 394: the format of this reference seems somewhat strange.
Author Response
Responses in italics.
The paper from Eberhart and co-workers which theoretically shows that bond-bundles can provide new and quantifiable insights into the structure and properties of molecules and solid-state compounds is a good work which was competently carried out and which can be published in Molecules. I have just a few minor matters that the authors should consider before publication proceeds.
-
- In addition to ‘merging the computational and synthetic branches of chemistry into a single subdiscipline’ to realize molecular design, AI may also be part of the story. This could be mentioned and referenced.
- A statement has been added in the sixth paragraph of the introduction: “Taken to larger scales, machine learning and artificial intelligence algorithms provide the upmost combination of computational empiricism and statistical analysis, and are increasingly used to better understand and predict molecular and material properties [cite].”
- A few bond descriptors in the real space have been developed and used over the years, see Chem. Theory Comput. 2014, 10, 3745 and refs therein for instance. They should be referenced and discussed with respect to the approach used in the paper.
- A reference to such real-space chemical bonding visualization and analysis methods was tentatively added in the second paragraph of section 2, though I prefer to not make negative statements in papers so I leave it to the editor’s discretion as to whether or not it should appear in the published manuscript: “While some real-space bonding representations are in use [cite], none of them include constructs possessing all four properties.”
- In addition to ‘merging the computational and synthetic branches of chemistry into a single subdiscipline’ to realize molecular design, AI may also be part of the story. This could be mentioned and referenced.
- The choice of silver as a solid-state compound – I prefer the word ‘solid-state compounds’ rather than ‘materials’ which connotates some application – and cyclobutadiene as examples should be justified.
- A justification has been added to the end of the introduction: “After developing the concepts, we include two examples using one solid state and one molecular system; pure silver and cyclobutadiene, respectively. The two provide contrast for organic vs metallic bond-bundles, while the cyclobutadiene example dissects a familiar chemical phenomenon, the Jahn-Teller distortion, including the assessment of individual bond energies.”
- Typo: Page 10, line 314. Thought
- Thank you.